A bike-sharing demand prediction model based on Spatio-Temporal Graph Convolutional Networks

Zhou Chaoran
Hu Jiahao
Zhang Xin zhangxin@cust.edu.cn
Li Zerui
Yang Kaicheng
School of Computer Science and Technology, Changchun University of Science and Technology , Changchun, Jilin , China
Piangerelli Marco
Electronic publication date: 2024 Oct 15
Publication date: 2024
Volume: 10
Electronic Location ID: e2391
Received 2024 Mar 12; Accepted 2024 Sep 16
Copyright: © 2024 Zhou et al.
Copyright year: 2024
Copyright holder: Zhou et al.
License: This is an open access article distributed under the terms of the Creative Commons Attribution License, which permits unrestricted use, distribution, reproduction and adaptation in any medium and for any purpose provided that it is properly attributed. For attribution, the original author(s), title, publication source (PeerJ Computer Science) and either DOI or URL of the article must be cited.
License URL: https://creativecommons.org/licenses/by/4.0/

Keywords: Demand prediction, Demand graph, Graph convolutional network (GCN), Spatio-temporal characteristic

Funding: Natural Science Foundation of Jilin Province 20200201182JC, 20210101474JC Natural Science Foundation of Jilin Province YDZJ202301ZYTS422 Science Research Foundation of Jilin Province JJKH20230846KJ Natural Science Foundation of Jilin Province 2021YFB1714400 The work is supported by the Natural Science Foundation of Jilin Province under grants 20200201182JC and 20210101474JC, the Natural Science Foundation of Jilin Province under grants YDZJ202301ZYTS422, the Science Research Foundation of Jilin Province under grants JJKH20230846KJ and the Natural Science Foundation of Jilin Province under grants the National key research and development program under grants 2021YFB1714400. The funders had no role in study design, data collection and analysis, decision to publish, or preparation of the manuscript.

==============================
Shared bikes, as an eco-friendly transport mode, facilitate short commutes for urban dwellers and help alleviate traffic. However, the prevalent station-based strategy for bike placements often overlooks urban zones, cycling patterns, and more, resulting in underutilized bikes. To address this, we introduce the Spatio-Temporal Bike-sharing Demand Prediction (ST-BDP) model, leveraging multi-source data and Spatio-Temporal Graph Convolutional Networks (STGCN). This model predicts spatial user demand for bikes between stations by constructing a spatial demand graph, accounting for geographical influences. For precision, ST-BDP integrates an attention-based graph convolutional network for station demand graph’s temporal-spatial features, and a sequential convolutional network for multi-source data (e.g., weather, time). In real dataset, experimental results show that ST-BDP has excellent performance with mean absolute error (MAE) = 1.62, mean absolute percentage error (MAPE) = 15.82%, symmetric mean absolute percentage error (SMAPE) = 16.14%, and root mean square error (RMSE) = 2.36, outperforming the baseline techniques. This highlights its predictive accuracy and potential to guide future bike-sharing policies.

Introduction

Currently, numerous cities globally are adopting bike-sharing (BS) systems at an expansive scale, integrating them seamlessly with established modes of public transportation such as buses and subways (Xu et al., 2019). This strategic amalgamation has been instrumental in mitigating the challenges associated with the last mile of commuting, traditionally a bottleneck in urban transportation networks (Fontugne, Shah & Cho, 2020; Fan, Chen & Wan, 2019). In recent years, the proliferation of BS initiatives has fostered positive urban development, not only facilitating convenient and efficient short-distance travel but also promoting energy conservation and environmental stewardship (Zhang & Mi, 2018; Wu, Kim & Chung, 2021; Cheng et al., 2021). Consequently, BS stands as a pivotal element in contemporary urban development, heralding a paradigm shift in transportation strategies focused on sustainability and efficiency (Zhu et al., 2022). As urban areas witness a surge in the utilization of BS systems (Lyu et al., 2021), the accessibility and convenience for users have notably escalated (Chen, van Lierop & Ettema, 2020).

However, this uptick has engendered a series of challenges, casting a shadow of adverse effects on societal dynamics (Wang, Huang & Dunford, 2019; Tekouabou, 2021). A prominent issue is the disparate spatial and temporal distribution of these bicycles, a phenomenon which exacerbated by divergent urban functionalities and user riding predilections. This has culminated in an over-saturation of bicycles in certain stations, while others face a scarcity, thereby escalating the operational expenditures associated with rebalancing the bicycle distribution. Presently, the predominant strategy employed by BS firms revolves around centralized transportation and drop-off systems to regulate the bicycle counts at various stations. This methodology, unfortunately, is marred by considerable spatial and temporal lags, an absence of foresight in deployment strategies, and a diminished capacity to cater to real-time user demands. Consequently, the imperative to foster research endeavors focusing on the prediction of users’ BS demand and strategic bicycle layout has become increasingly salient.

Conventional approaches to forecasting demand in BS systems often neglected the critical spatio-temporal dimensions, resulting in predictions characterized by low accuracy due to the insufficient incorporation of the features such as weather, time signal, stations’ location relationship. Furthermore, overlooking the intrinsic interconnections among spatial network stations engenders resource misallocation in BS deployments. To address these significant shortcomings inherent in traditional methodologies, we propose the ST-BDP model. This innovative work is designed to offer the following key contributions:

1) The input for ST-BDP adeptly integrates features pertaining to spatio-temporal information, encompassing aspects such as weather conditions, time series signals, and the demand characteristics of space station groups. In constructing the demand characteristics of the space station groups, the concept of “Tobler’s First Law of Geography” (Tobler, 1970) is integrated. This is achieved by weighting the influence of station-location relationships through the station group demand map at various moments, thereby facilitating the fusion of spatial relationship information between stations and demand relationship information at different time points.

2) ST-BDP’s novel structure ensures the refined extraction and utilization of spatio-temporal information for predictive analysis. The encoder utilizes a multi-input feature structure to precisely extract information from various types of spatio-temporal features. In the decoder segment, aiming for an accurate prediction of the varying demand trends for bike-sharing at each station, we have incorporated a Self-Attention computation layer to meticulously extract the encoded feature information.

3) A demonstration that ST-BDP convincingly outperforms methodologies such as STGCN (Yu, Yin & Zhu, 2018) and GCGRU (Li et al., 2018). Compared to the baseline works, ST-BDP exhibits the best performance in terms of MAE, MAPE, SMAPE, and RMSE. Owing to the specialized design of the ST-Conv and TCN blocks, ST-BDP demonstrates exceptional performance in balancing time consumption and parameter configuration, all while capturing and mining the spatio-temporal features from multiple data sources.

This article is organized as follows. “Related Work” provides a comprehensive analysis of related research works. “ST-BDP Model” describes the data feature extraction and model details. “Experiment” provides database information, experimental parameters, and experimental results. “Discussions” discusses the experimental results and analyzes the novelty of the model. “Conclusions” concludes the article.

Related work

Predicting the demand for bike-sharing is integral to the broader field of traffic flow forecasting (Jiang & Luo, 2022). Currently, three principal methodologies have been employed to address the task of predicting bike-sharing user demand. These methodologies encompass time-series prediction (Singhvi et al., 2015; Cheng et al., 2020; Zhao et al., 2022; Zhu, 2022), approaches grounded in machine learning (Albuquerque, Sales Dias & Bacao, 2021), and those utilizing deep learning techniques (Jiang, 2022).

Traditionally, researchers in this field have primarily leveraged time-series modeling methods to forecast patterns and fluctuations in traffic flow. Singhvi et al. (2015) introduced a logistic regression model designed to predict bicycle demand during weekday morning peak hours (7:00 a.m.–11:00 a.m.) in New York city. This model incorporated considerations of cabs, weather, and spatial factors to enhance prediction accuracy. Cheng et al. (2020) established a stochastic polynomial logit model, integrating sets of polynomial logit models, to forecast the travel demand of urban tourists. Sathishkumar, Park & Cho (2020) propose a data mining-based approach to forecast hourly bicycle rental demand, and train the best performing algorithm with a combination of various features. Zhao et al. (2022) addressed the seasonal traffic forecasting problem by employing autoregressive integrated moving average-maximum correntropy criterion (ARIMA-MCC) and conditional kernel density estimation-generalized autoregressive conditional heteroscedasticity (CKDE-GARCH) for forecasting urban short-term traffic flow. Zhu et al. (2022) formulated a multi-objective mixed integer linear programming method, accounting for multiple influencing factors, to predict the route problems encountered by bicycle tourists. In recent years, traditional time-series prediction methods such as Historical Average (HA) (El Esawey, 2018) and Autoregressive Integrated Moving Average model (ARIMA) (Yoon, Pinelli & Calabrese, 2012; Ospina et al., 2023) have been utilized for traffic flow forecasting, primarily serving as baseline models for comparison with machine learning and deep learning models.

In recent decades, machine learning-based approaches have witnessed significant advancements in various domains, including traffic flow forecasting and bike-sharing demand prediction. Li et al. (2015) developed a dichotomous clustering algorithm and an XGBoost-based system for predicting a city’s total bike rentals. Guido, Rafał & Constantinos (2019) pioneered a unique, low-dimensional method that integrates public and weather data, using clustering techniques to reduce dimensionality and requiring minimal assumptions for accurate demand forecasts. Ashqar, Elhenawy & Rakha (2019) utilized random forest and bootstrap forward stepwise regression to forecast the number of stations in a bike-sharing system, incorporating weather conditions. Xu et al. (2020) introduced a hybrid model using edge computing and machine learning, which fuses self-organizing mapping networks and regression trees to efficiently predict the demand for shared bikes at a single site. Sathishkumar & Cho (2020) compare the performance of CUBIST, regularized random forest, classification and regression trees, k-nearest neighbours and conditional inference tree, and then a rule-based regression prediction model was proposed to predict the demand for shared bicycles. Parsa et al. (2021) applied three machine learning techniques—k-nearest neighbor (KNN), random forest (RF), and extreme gradient boosting (XGBoost)—to model network traffic flow predictions. By introducing random forest model, combining annual and seasonal data and analyzing the importance of different variables, Sathishkumar & Cho (2024) establish prediction demand models for shared bicycles in different seasons. These machine learning-based approaches demonstrate notable efficacy; however, when dealing with spatio-temporal traffic data exhibits big data characteristics, there lies a potential risk of inadequately uncovering obscure features (Bouktif et al., 2018).

With the progressive advancement of deep learning techniques, their application in time series prediction has become increasingly prevalent. The ability of deep learning methodologies to adeptly decipher complex patterns and relationships within datasets underscores their growing adoption in diverse predictive scenarios. RNN-based deep learning models, such as recurrent neural network (RNN) (Elman, 1990), long short-term memory (LSTM) (Graves, 2012) and gated recurrent unit (GRU) (Cho et al., 2014), have been extensively employed for the prediction of time series data. Wang & Kim (2018) utilized LSTM and GRU to develop efficient models for predicting shared bike availability at stations, demonstrating their intricate structures’ effectiveness. Concurrently, Collini, Nesi & Pantaleo (2021) crafted a dynamic LSTM-based model proficient in forecasting short-term bike demand at intervals of 15 to 60 min. In 2020, Xu, Liu & Yang (2020) proposed a novel multi-block hybrid model, which, by acknowledging both spatio-temporal characteristics and external dependencies, leveraged multiple data sources to precisely predict bike-sharing supply and demand dynamics. Additionally, research has underscored that while RNN-based models, inherent in their sequential nature, showcase a robust ability to mine temporal features, they exhibit diminished computational efficiency; conversely, CNNs, renowned for their proficiency in parallel computation, grapple with constraints in capturing temporal features (Goswami & Kumar, 2022). Seeking to overcome existing model limitations, Bai, Kolter & Koltun (2018) introduced the temporal convolutional network (TCN) for sequential data. By incorporating causal and dilated convolutions into CNN, TCN merges CNN’s parallel computation benefits with improved temporal feature capture, displaying advantages such as parallelism, stable gradients, a flexible receptive field, and reduced memory usage over RNN models. Despite its significant advancement in temporal prediction accuracy compared to CNN, TCN still encounters challenges in capturing spatial features adequately.

In the realm of exploring spatial feature learning, graph neural networks (GNNs) have demonstrated exceptional proficiency in delving into the graph structural information inherent in traffic road networks and other applications (Zhou et al., 2020; Wu et al., 2020). Lin, He & Peeta (2018) introduced a novel model leveraging GNN, incorporating a data-driven graph filter designed to discern the hidden correlations between stations, aiming to predict hourly demand at the station level within a shared-vehicle network. The STGCN (Yu, Yin & Zhu, 2018) framework, rooted in GNN methodologies, was developed to adeptly address the intricacies of traffic forecasting and has demonstrated superior performance. Addressing the non-periodic nature of taxi demand variations and the temporal shifts in spatial correlations, Yao et al. (2019) proposed a spatio-temporal dynamic network (STDN). This innovative model incorporates a gating mechanism and an attention mechanism with periodic shifts, enabling it to learn dynamic similarities between locations and adapt to temporal shifts over extended cycles. Guo et al. (2019) proposed an attention based spatial-temporal graph convolutional network (ASTGCN) model to capture spatial-temporal traffic information and solve forecasting problem. Presently, a burgeoning number of sophisticated models based on GNNs have been consecutively proposed and effectively employed to address a variety of issues pertaining to traffic flow forecasting (Jiang & Luo, 2022; Xu et al., 2023; Lian et al., 2023).

The above research results are partially shown in Table 1, the research results illustrate that both TCN (Bai, Kolter & Koltun, 2018) and STGCN (Yu, Yin & Zhu, 2018) not only inherit the parallel computational advantages of CNN but also rival RNNs in learning temporal feature, demonstrating their exemplary spatio-temporal feature extraction (FR) capabilities. Moreover, in the context of traffic forecasting (Pavlyuk, 2019) and broader research scenarios (Tang et al., 2022), appropriate input feature selection can substantially enhance model learning performance. In light of these findings, we have incorporated weather conditions, time series signals, and the demand characteristics of space station groups as input features for ST-BDP model, and subsequently validated the efficacy of such feature selection through rigorous experimentation.

Table 1 The summary of work related to shared bikes.

Method	Reference	Model	Dataset	Performance	
Time-series prediction	Singhvi et al. (2015)	Regression analysis	Citi Bike’s website	RMSE = 163 (an absolute scale)	
	Zhu et al. (2022)	Multi-objective optimization	Trip Advisor 2019	–	
Machine learning	Li et al. (2015)	Bipartite clustering algorithm Gradient Boosting Regression Tree Multi-similarity-based inference model	Capital Bike share system Citi Bike system	0.03 reduction of error rate	
	Guido, Rafał & Constantinos (2019)	Aggregation and Clustering Prediction and Disaggregation	New York City bike system	After decomposition,the RMSE of average prediction error decreased 40	
	Ashqar, Elhenawy & Rakha (2019)	Random Forest Regression Model Bayesian Information Criterion	Anonymized bike trip data	MPE: 1.1–2.3	
	Xu et al. (2020)	Self-organizing mapping network Regression tree	Washington and London bike-sharing systems	MAE = 61.073 RMSE = 62.650 RMSLE = 0.068	
Deep learning techniques	Wang & Kim (2018)	RF LSTM GRU	Suzhou Youon Public Bicycle Systems	Sequence Length = 30 Time Interval = 10 Station 628; MAE:LSTM 1.3-1.4, GRU 1.2-1.3, RF 1.2-1.3	
	Collini, Nesi & Pantaleo (2021)	Bi-LSTM	Data collected in bike-stations in the cities of Siena and Pisa (Italy)	MAE: 0.5–1.0	
	Xu, Liu & Yang (2020)	MBH Model (CNN GRU-Net ConvGRU-Net)	Spatial-temporal characteristics of GPS data in Shanghai	Acc. = 91.51%	

St-bdp model

This chapter provides an in-depth description of the ST-BDP model’s functional structure with specifically emphasis on its spatio-temporal feature construction. Additionally, the functions and roles of the multi-input-based Encoder and the attention-driven Decoder components are comprehensively introduced.

Spatio-temporal feature construction

The input spatio-temporal feature of ST-BDP includes weather conditions, time series signals and demand characteristics of space station groups.

Demand graph feature

Demand graph feature description

The bike-sharing user demand at various bike-sharing stations in the cities is interdependent. This phenomenon aligns with “Tobler’s First Law” (TFL), which posits that “All things are related to other things, but near things are more related than distant things” (Tobler, 1970). To capture the intricate relationships between bike-sharing stations, the historical demand characteristics are structured into graph features, encapsulating the influence dynamics among stations. By introducing demand graph features, ST-BDP can have the ability to accurately identify the spatio-temporal information in historical demand data.

In the demand graph feature for bike-sharing stations, each station represents a vertex, while the physical distance between stations defines the edges. The weights of the edges represent the influence of users on the demand for shared bikes between stations. The magnitude of bike demand at each station is intrinsically linked to these edges, facilitating the construction of demand graph characteristics for the bike-sharing stations.

The graph features are defined as G=(V,E,W), where V is the vertex of the graph, E is the vertex of the graph and the edge between the vertices, and the relationship between the individual nodes will form a N×N-dimensional matrix, i.e., the adjacency matrix W, W∈RN×N. In the demand graph feature for bike-sharing stations, each vertex value represents the demand at a specific station at a given time. Let T denote the time interval, with each interval Ti corresponding to a specific demand map. Figure 1 illustrates the demand graph across various time periods. At any given time Ti, the graph can be represented as GTi=(VTi,E,W); here, VTi is the vertex set indicating the demand at n bike-sharing stations. E denotes the edge set, where each edge signifies the relationship between two interconnected stations. Meanwhile, W∈RN×N stands for the adjacency matrix.

Figure 1 Demand graph for bike-sharing stations.

The weights of the adjacency matrix W are computed using Eq. (1). In this equation, dij represents the distance between stations i and j. wij signifies the weights associated with the edges in set E. σ2 corresponds to the distribution variance of the matrix W, and θ is the sparsity threshold value.

The demand graph features of bike-sharing stations are constructed to reflect the relationship between the station locations and their corresponding users’ demand.

(1) wij={exp⁡(−dij2σ2),i≠jandexp⁡(−dij2σ2)≥θ0,others.

Demand graph feature construction

We construct demand graph features using shared bike riding data from users in New York City, USA from 2017/01/01 to 2020/12/31. This data was sourced from the Citi Bike sharing system, accessible at https://www.citibikenyc.com/system-data. Table 2 shows the data items of bike-sharing riding data.

Table 2 The data items of bike-sharing riding data.

Data items	Data type	Unit	
Start time	Int	second	
Start station ID	String	–	
Start station name	String	–	
Start station latitude	String	–	
Start station longitude	String	–	
End time	Int	second	
End station ID	String	–	
End station name	String	–	
End station latitude	String	–	
End station longitude	String	–	

Using the riding data, the hourly usage of bike-sharing at each station is computed and consolidated into a dataset representing users’ bike-sharing demand. Examples of demand data for various stations are presented in Table 3.

Table 3 Examples of bike-sharing demand data.

Time	Station A	Station B	Station C	Station D	Station E	
2019-07-09 06:00	20	5	5	8	10	
2019-07-09 07:00	48	19	18	22	20	
2019-07-09 08:00	106	26	26	28	32	
2019-07-09 09:00	68	30	22	26	27	
2019-07-09 10:00	13	20	11	18	15	
2019-07-09 11:00	20	21	15	15	15	
2019-07-09 12:00	14	27	19	19	19	
2019-07-09 13:00	13	21	19	19	16	
2019-07-09 14:00	16	15	23	24	21	
2019-07-09 15:00	21	33	22	18	15	
2019-07-09 16:00	51	27	30	40	29	
2019-07-09 17:00	163	78	111	47	65	
2019-07-09 18:00	108	67	137	41	88	

To prevent the data from entering the saturation region of the activation function and subsequently causing vanishing gradients, we normalize the demand data. Given that the primary activation functions of the models are the Tanh and ReLU function—which are particularly sensitive to values within the range [−1,1]—the demand data is normalized to fall within this range, thereby facilitating more effective model training.

The adjacency matrix of the bike-sharing demand graph feature is calculated by the distance between the bike-sharing stations in the traffic network, and the calculation (Eq. (1)). The time interval of the data set is set to 1 h, and each hour corresponds to one bike-sharing demand graph, i.e., each vertex in the network has 24 data points per day.

Weather feature

Weather feature description

Natural weather determines whether users choose bike sharing as a mode of travel, which in turn affects the demand for bike sharing. Therefore, in order to make accurate demand forecasts for shared bikes, weather factors need to be considered. Combined with spatio-temporal feature equations, a series of processes of data attribute calibration, cleaning, feature encoding and scaling are required in order to extract weather features from the data.

Attribute calibration and data cleansing. The primary attributes of weather data encompass time stamps, temperature, dew point, humidity, and wind speed. These data, collected at regular intervals, are associated with weather indicators for that time period. Due to collection irregularities, there are occasional redundancies and missing values. For integer-type data, linear interpolation is employed for imputation, while string-type attributes such as wind direction and weather type utilize up-filling techniques.

Feature coding. Weather-related data attributes, such as wind direction and weather type, are typically represented as strings. To facilitate their use in ST-BDP, certain feature data undergoes encoding and transformation processes. Specifically, the fixed number of weather types is represented by using one-hot encoding. Meanwhile, string-based wind data is translated into an angular representation, delineating wind direction within the [0, 360] degree range, with 0 degrees denoting due north and encompassing 16 distinct wind directions. Notably, direct angle values present challenges for model inputs, especially when values approach both 0 and 360 degrees, which should be interpreted as closely adjacent. To address this, wind direction and speed are translated into wind vectors, as demonstrated in Eqs. (2) and (3).

(2) windx=speed×cos⁡(direction×π180)

(3) windy=speed×sin⁡(direction×π180).

Feature scaling. Weather data exhibits a long-tailed distribution, where samples are unevenly distributed, and there’s a substantial variance in data values. This imbalance can adversely affect subsequent ST-BDP training and degrade the overall model performance. To mitigate this, we introduce Eq. (4), a feature normalization approach for weather attributes. In the equation, x represents the input feature, xmean denotes its mean, and xstd stands for its variance. By employing this normalization, we can alleviate issues arising from the skewed sample distribution, harmonize feature value disparities, and markedly enhance the stability of the quality of ST-BDP’s input weather feature.

(4) y=x−xmeanxstd.

Weather feature construction

The weather data of LaGuardia Airport in New York, USA was selected to construct the weather feature of ST-BDP for experiments. Table 4 shows the data type and unit of the main data items of the weather data, all from the Weather Underground website (https://www.wunderground.com).

Table 4 Data items of weather data.

Data items	Data type	Unit	
Timestamp	Int	second	
Temperature	Int	F	
Dew point	Int	F	
Humidity	Int	%	
Wind speed	Int	mph	
Pressure	Float	in	
Precipitation	Float	in	
Wind direction	String	–	
Condition	String	–	

Weather data undergoes data cleaning and encoding procedures to transform wind direction and wind speed into wind vectors. Figure 2A depicts the wind direction and wind speed prior to conversion, and Fig. 2B illustrates the resultant wind vectors post-conversion.

Figure 2 Wind and wind vectors.

To mitigate the long-tail distribution and achieve a more uniform data spread, the weather feature within the dataset is normalized. Figure 3 displays the violin plots of primary weather attributes both prior to and following the feature scaling process.

Figure 3 Violin plots of weather data.

Table 5 shows some example results of the weather data used in experiment after feature scaling.

Table 5 Examples of weather features (T: Temperature; D: Dew point; H: Humidity; W X: Wind vector X; W Y: Wind vector Y).

Time	T	D	H	W X	W Y	
06:00	72	65	78	2.583	−5.4156	
07:00	72	65	78	3.536	−3.536	
08:00	74	65	73	5	0	
09:00	78	66	66	1.148	2.772	
10:00	81	63	54	0	0	
11:00	83	59	44	6.364	−6.364	

Upon compression of the weather features, a data analysis was conducted, followed by a linear regression with the demand for bike-sharing. As depicted in Fig. 4, there exists a positive correlation between the demand for bike-sharing and both temperature and dew point. Conversely, humidity and precipitation exhibit a negative correlation with demand. The relationships between demand and wind speed, as well as between demand and air pressure, are not clearly delineated and cannot be succinctly represented as linear associations. This also proves the rationality of ST-BDP model using deep learning methods to learn these nonlinear complex relationships.

Figure 4 Demand data distribution under different weather features.

Temporal feature

Temporal feature description

Bike-sharing demand exhibits periodic patterns across daily, weekly, and yearly timeframes. Nonetheless, representing time in mere seconds results in a monotonic increase, failing to capture these inherent cyclical tendencies as indicated by Phillips, Parr & Riskin (2003). In light of this, temporal features are constructed using sine and cosine functions, capitalizing on their intrinsic periodic properties to effectively represent temporal data.

Time t in seconds is extracted from the weather and bike-sharing demand data. t is initially transformed into a timestamp, which is then further converted into time signals—specifically, sine and cosine signals. The employed time signals encompass daysin, daycos, weeksin, weekcos, yearsin, and yearcos. These signals capture the periodicities in bike-sharing demand across varying durations: daily, weekly, and yearly. Serving as crucial temporal characteristic, these signals constitute one of the input features for the ST-BDP model.

Equations (5) and (6) facilitate the transformation of daily timestamps into daysin and daycos. For every hour, a unique temporal point is denoted by the pair (daysin,daycos). This pairing effectively represents a specific hour within a day, with both daysin and daycos completing a cycle over a 24-h span, thereby encapsulating the daily periodicity.

(5) daysin=sin⁡[timestamp×(2×π60×60×24)]

(6) daycos=cos⁡[timestamp×(2×π60×60×24)]

Equations (7) and (8) facilitate the transformation of weekly timestamps into weeksin and weekcos. These functions, weeksin and weekcos, complete one cycle over a span of 1 week (equivalent to 7×24 h), capturing the weekly periodicity exhibited by the demand for bike-sharing.

(7) weeksin=sin⁡[timestamp×(2×π60×60×24×7)]

(8) weekcos=cos⁡[timestamp×(2×π60×60×24×7)]

Equations (9) and (10) are employed to transform yearly timestamps into yearsin and yearcos. These terms, yearsin and yearcos, cycle annually, with each instance within the year being uniquely represented by the coordinate pair (yearsin,yearcos). They encapsulate the annual periodicity inherent in the demand for bike-sharing.

(9) yearsin=sin⁡[timestamp×(2×π60×60×24×365.2425)]

(10) yearcos=cos⁡[timestamp×(2×π60×60×24×365.2425)].

Temporal feature distribution

Daily demand data distribution. The demand for bike-sharing is related to time and exhibits distribution characteristics based on daily, weekly, and yearly cycles. Taking the Pershing Square North station (with a longitude of 40.751873 and a latitude of −73.977706) from the dataset used in this study as an example, its relationship between bike-sharing demand and time within 1 day (from 0–23 h on 2017/2/13) is shown in Fig. 5A. Similarly, the relationship between bike-sharing usage and time over a week (from 0–168 h, between 2017/2/13 and 2017/2/19) is depicted in Fig. 5B. Within a single day, there are two peak demand periods for shared bicycles, which correspond to the morning rush hours from 7:00 to 9:00 and the evening rush hours from 17:00 to 19:00. The usage at night is lower than during the day, and this pattern is consistent every day. This indicates that the demand for bike-sharing follows a daily cyclical distribution pattern.

Figure 5 Time signal (day).

Weekly demand data distribution. The demand for bike-sharing exhibits a cyclical pattern on a weekly basis. Taking the Pershing Square North station as an example, its line chart of demand over three weeks (from 0–504 h, between 2017/2/13 and 2017/3/6) is shown in Fig. 6A. The results of grouping and summing the demand for bike-sharing by day of the week are presented in Fig. 6B. The demand for bike-sharing on weekends is noticeably lower than on weekdays, and this pattern of demand distribution is consistent each week. This illustrates that the demand for bike-sharing follows a weekly cyclical distribution pattern.

Figure 6 Time signal (week).

Yearly demand data distribution. The demand for bike-sharing exhibits an annual periodicity. Taking the Pershing Square North station as an example, the relationship between the demand for bike-sharing usage and time over three years is shown in Fig. 7. It can be observed that the bike-sharing usage demand has a similar distribution pattern each year, indicating that the demand for bike-sharing follows an annual cyclical pattern. Furthermore, from December to February of the following year, which constitutes winter in New York, there’s a noticeable dip in the demand for bike-sharing. This highlights that weather conditions also have a significant impact on the long-term demand for bike-sharing.

Figure 7 Time signal (year).

According to the previous description, the demand for bike-sharing shows a distribution pattern with daily, weekly, and yearly cycles. For the construction of temporal features, the data are processed by sine and cosine functions (Eqs. (5), (6)) which are used to represent the temporal feature information of temporal data. Figure 8 shows the day signal plot for a day, with the horizontal coordinates of time and the vertical coordinates of the values of daysin and daycos. daysin and daycos characterize the periodicity of the demand for bike-sharing usage in terms of days.

Figure 8 Time signal (day).

Equations (7)–(10) are utilized to convert the timestamp into weeksin and weekcos (the period is 7 × 24 h), and yearsin and yearcos (the period is 365 days) to construct temporal features with time periodicity. Table 6 shows some of the time signals in the study data.

Table 6 Time signals.

Time	Day sin	Day cos	Year sin	Year cos	
06:00	−0.5	0.866	−0.110	−0.99392	
07:00	−0.259	0.966	−0.111	−0.99384	
08:00	−0.751	1	−0.112	−0.99376	
09:00	0.259	0.966	−0.112	−0.99368	
10:00	0.5	0.866	−0.113	−0.9936	
11:00	0.707	0.707	−0.114	−0.11366	

Model structure

ST-BDP model operates on an encoder-decoder architectural framework. Within ST-BDP model, the STGCNTCN and STGCN blocks are adeptly employed to distill diverse spatio-temporal characteristics, thereby enhancing the accuracy for bike-sharing demand predictions. The ST-BDP model, architected in an Encoder-Decoder framework, predominantly encompasses four neural networks: STGCN, TCN, Conv1D, and Self-Attention. The structure of ST-BDP model is depicted in Fig. 9.

Figure 9 ST-BDP model structure diagrams.

Spatio-temporal feature construction: The multi-input for ST-BDP adeptly integrates features pertaining to spatio-temporal information, encompassing aspects such as demand graph feature of space station groups (demand-characteristic), weather conditions (weather-characteristic), and time series signals (time-characteristic).

STGCN and TCN-based encoder: ST-BDP’s encoder is designed for feature vector encoding. The TCN blocks are utilized to extract the time-characteristic and weather-characteristic’s feature. The GNN-based STGCN block is utilized to extract demand characteristics’ spatio-temporal feature. Finally, the comprehensive representation of all input features is obtained through the encoder’s feature fusion layer, Con1D, residual cascade, and normalization layer.

Self-Attention-based decoder: The feature vector output by Encoder is transformed into time series data of future time period by Self-Attention and residual connection, and finally dimensional transformation is performed by Conv1D to achieve the prediction of future demand of bike-sharing.

Relative to conventional demand forecasting frameworks, ST-BDP model based on STGCN has the advantage of richer training parameters and a deeper model. Notably, all layers of ST-BDP model are convolutional operations or other structures that can process in parallel. This model structure, devoid of sequential operations, offers a distinct advantage in training time over RNN-based configurations. Leveraging the ResNet network architecture (Targ, Almeida & Lyman, 2016), the Add block within ST-BDP model integrates the dense vector produced by the feature fusion layer with the output vector from the STGCN Block. This integration ensures that the demand characteristics of the spatial station clusters significantly influence the prediction results, while mitigating potential interference from secondary features. Concurrently, this approach addresses the vanishing gradient issue in ST-BDP model, which can arise due to an excessive number of neural network layers.

STGCN and TCN-based FR block

In ST-BDP model, the Encoder component is responsible for encoding feature vector, which is subsequently processed by a feature fusion layer and a residual concatenation layer for enhanced feature learning. This component is composed of modules: the STGCN block, the TCN blocks, feature fusion layer, residual concatenation layer, and a normalization layer. Specifically, the STGCN block focuses on extracting hidden information from the demand graph features, while the TCN blocks are dedicated to capturing temporal and weather-related feature information.

TCN block

Block2 and Block3, which are utilized to extract weather and temporal features, are composed of four stacked TCN residual blocks (Bai, Kolter & Koltun, 2018). The architecture of a TCN residual block is illustrated in Fig. 10. A TCN residual block encompasses a series of transformations. The output of TCN block is obtained by adding its input x to the transformations F(x) made within the residual block, as represented in Eq. (11).

(11) o=Activation(x+F(x)).

Figure 10 The structure of TCN block.

The TCN residual block is composed of two dilated convolution layers (Yu & Koltun, 2015) and a ReLU layer. Weight normalization (Salimans & Kingma, 2016) is employed for normalization. Moreover, a dropout layer (Srivastava et al., 2014) is added after each dilated convolution for regularization purposes. To accommodate varying input and output dimension sizes, beyond the basic 3×3 convolution kernel, an additional 1×1 convolution kernel is utilized for dimension transformation. This transforms the input into a tensor with the same shape as the output, ensuring that the input and output can be added together seamlessly.

STGCN block

STGCN block, as shown in Fig. 11A, is a three-layer structure consisting of two spatio-temporal convolutional blocks (ST-Conv blocks) and one output block. This structure also facilitates the network to fully utilize the bottleneck strategy, achieving dimension compression and feature compression, and adjusting the channel count through the graph convolution layer. Figure 11B illustrates that the ST-Conv block possesses a three-layered structure. The first and third layers are direction temporal gated-Conv layers (Fig. 11C), the second layer is a spatial Graph-Conv layer, the output layer is a convolutional layer, and residual connections and bottleneck strategies are introduced in each block. The input vt−M+1,...,vt is uniformly processed by ST-Conv block to capture the spatio-temporal dependencies of the data. Finally, the output layer integrates all features to generate the final prediction result v^.

Figure 11 The structure of STGCN block.

The input and output of the ST-Conv block are both 3D tensors. The input of block l is vl∈RM×n×Cl, and the output is vl+1∈R(M−2(Kt−1))×n×Cl+1. Refer to Eq. (12) for further details.

(12) vl+1=Γ1l×τReLU(Θl×G(Γ0l×τvl))

Γ1l and Γ0l represent the convolution kernels of the upper and lower Temporal Gated-convs layers of block l, Γ0l represent the convolution kernel of the spatial Graph-Conv layer, ReLU() represent the ReLU activation function. The output block maps the output of the last ST-Conv block into a matrix representing the final prediction result.

The loss function of STGCN is shown in Eq. (13), where Wθ is all the trainable parameters in the model, vt+1 is the real label, and v^() represents the predicted value of the model.

(13) L(v^,Wθ)=∑t||v^(vt−M+1,...,vt,Wθ)−vt+1||2.

The remaining encoder layers

ST-GCN Block1, TCN Block2, and TCN Block3 produce three encoded vectors x1, x2, and x3 corresponding to the input features. These vectors x1, x2, and x3 are merged into a dense vector x^ through a feature fusion layer, comprising a concatenate layer followed by a Conv1D layer. The Concatenate layer stitches the x1, x2, and x3 vector into a vector x^, and Conv1D is used to realize the feature fusion and vector dimension transformation.

When passing through the feature fusion layer, vectors influence each other, potentially resulting in the loss of some important features. In the ST-BDP model, we introduce the residual connection in the ResNet (Targ, Almeida & Lyman, 2016) network and combine the dense vector output of the feature fusion layer with the vector output of STGCN Block1 through the Add block. This ensures that the historical demand feature of bike-sharing significantly influences the prediction result, safeguarding it from interference by less important features. Additionally, this approach prevents the gradient vanishing issue that might arise in the ST-BDP model due to an excessive number of neural network layers.

After multiple layers of processing in ST-BDP, data may deviate from normalization and exhibit an increasing bias. To ensure consistent training behavior, it is pivotal to re-normalize the data such that it exhibits a mean of 0 and a variance of 1. This is especially crucial in prediction task where the spatio-temporal data is diverse in each batch. Given the inherent variability in the data sets, these data may significantly diverge from each other, resulting in minimal correlation between them. To address these challenges, a normalization layer is employed for ST-BDP construction. For a given hidden layer l, with H representing the number of hidden neurons, and a being the pre-activation value of a neuron (i.e., a=wx), the expectation and standard deviation for normalization are derived as per Eqs. (14) and (15), respectively.

(14) ul=1H∑i=1Hail

(15) σl=1H∑i=1H(ail−ul)2.

Subsequently, the normalized output is computed using Eq. (16), where parameters g and b signify the gain and bias. Importantly, these parameters are adaptable and are fine-tuned during the training process. This normalization ensures that the activations across the features in a layer maintain consistent statistics, facilitating a stable and efficient training regimen.

(16) h=Relu(glσl.(ail−ul)+b).

The described structure constitutes ST-BDP’s encoder layers. These layers of the encoder process all inputs and encode them into a feature vector that captures the spatio-temporal characteristics.

Self-attention-based decoder

ST-BDP’s decoder is tasked with interpreting the vectorized output from the Encoder layers, ultimately yielding the projected bike-sharing demand for subsequent time frames. It encompasses a variety of blocks including Self-Attention computing, residual connections, one-dimensional convolutional networks, and normalization.

The decoder of ST-BDP is implemented based on the Self-Attention mechanism. Distinct from RNN-like structures that rely on sequential computations, Self-Attention operates in parallel, thereby offering superior computational efficiency. This mechanism is adept at capturing long-term dependencies without being hampered by the temporal distance between events. Moreover, Self-Attention prioritizes salient features and diminishes the effect of less pertinent ones on the predictions, an attribute that bolsters the precision of the predicted outcomes.

The Self-Attention computation within ST-BDP model is a transformative process where a vector sequence X undergoes encoding to produce another vector sequence Y, and the entirety of this Self-Attention computing is executed in parallel. The Scaled Dot-Product Attention mechanism is given by Eq. (17), where dk represents the dimension of k.

(17) Attention(Q,K,V)=softmax(QKTdk)V.

The Self-Attention algorithm employs the aforementioned Scaled Dot-Product Attention formula. For Self-Attention, Q, K, and V all emanate from a unified vector. Specifically, within the ST-BDP model, Q, K, and V are derived by the multiplication of the Encoder’s output vector x with three distinct matrices: w1,w2, and w3. This process is illustrated in Eq. (18). Notably, matrices w1,w2, and w3 undergo training in tandem with ST-BDP model’s other hyper parameters.

(18) Q,K,V=w1(x),w2(x),w3(x).

After undergoing the Self-Attention layer, the output vector x is transformed into another vector y. This vector y is then combined with x to produce the resultant vector y~. To expedite the training process, y~ is processed through a Layer Normalization layer. Subsequently, a one-dimensional convolution, Conv1D, is employed to convert the feature dimensions and deliver the predict of future bike-sharing demand. Opting for Conv1D over a linear layer was a strategic choice, as the convolutional operation intrinsic to Conv1D offers fewer parameters, resulting in more efficient training. The Decoder culminates its operations by producing a matrix symbolizing future bike-sharing demand.

Experiment

Dataset description

The bike-sharing demand dataset comprises demand graph data, weather data, and temporal data. Concerning temporal configurations, both input and output time spans are fixed at 72 h. Consequently, data pertaining to demand, weather specifics, and temporal signals from the preceding 72 h are employed to predict bike-sharing demand for the subsequent 72 h. This equates to a sliding window of 144 h, where the initial 72 h of demand data, weather information, and time signals serve as inputs to ST-BDP model, while the demand for the ensuing 72 h acts as labels. This window is then shifted backward hourly until the entirety of the dataset is processed.

Experimental settings

Experimental environment and model training

The experimental setup employed for the program was configured within the PyCharm IDE, utilizing Tensorflow 2.4 as the deep learning development library and Jupyter Notebook as the development tool. The hardware specifications of the experimental equipment include an Intel(R) Core(TM) i5-9300H CPU @ 2.40 GHz processor, an NVIDIA GeForce GTX 1660 Ti graphics card, and 16GB of DDR4 memory.

For model training, the ST-BDP model employed the Adam optimizer, set with a learning rate of For model training, the ST-BDP model employed the Adam optimizer, set with a learning rate of triggered when the decrease in the loss value is less 19, with its functional representation illustrated in Fig. 12. Given the susceptibility of bike-sharing demand predictions to anomalies—such as special holidays or extreme weather events which can induce unpredictable demand fluctuations—the Log-Cosh function was deemed suitable for this task. This function is twice differentiable everywhere and effectively mitigates the impact of outliers during model training.

(19) Loss=∑i=1mlog⁡(cosh⁡(yi−y^i)).

Figure 12 Function image of Log-Cosh.

For the ST-BDP model, the following parameter configurations were employed:

STGCN component:

The ST-Conv block, which is a part of the STGCN, has three layers with channels set to 64, 16, and 64, in sequential order.

Both the graph convolution kernel size (K) and the temporal convolution kernel size ( Kt) are configured to 3.

TCN component:

The TCN block within the ST-BDP model comprises four stacked TCN residual blocks. All these blocks utilize convolutional kernels of size 3.

Training details:

Optimizer: Adam optimizer is employed for the model training.

Loss function: The chosen loss function is Log-Cosh.

Time slice: The length of the time slice is set to 72.

Batch size: For each iteration, the batch size is fixed at 50.

Initial learning rate: Set to 0.001.

Early stopping criterion: Training is terminated prematurely if the loss value descends below 0.0002 for three consecutive times.

Model evaluation metrics

To evaluate the models’ performance, mean absolute error (MAE), mean absolute percentage error (MAPE), symmetric mean absolute percentage error (SMAPE), and root mean square error (RMSE) are selected as the evaluation metrics of the model.

(20) MAE=1m∑i=1m|yi−y^i|

(21) MAPE=100%M∑i=1m|yi−y^iy^i|

(22) SMAPE=100%M∑i=1m|yi−y^i|(|yi|+|y^i|)/2

(23) RMSE=1m∑i=1m(yi−y^i)2.

Equations (20)–(23) are the MAE, MAPE, SMAPE, and RMSE calculation formulas, respectively, where m denotes the total number of test samples, while yi and y^i denote the actual and predicted values, respectively.

The influence of various saptio-temporal features

To observe the influence of various features on the prediction results, we constructed a single-layer linear computational model, taking historical demand, weather, and time data as inputs to predict the future usage demand for bike-sharing. By analyzing the model parameters, we derived the influence weight ratio of input features on the output, as shown in Fig. 13. The experimental results demonstrate that the absolute value of the influence weight of historical demand exceeds 0.6, making it the most dominant feature among all the input characteristics. Thus, historical demand is the most influential factor on the prediction results, indicating that it is the most crucial input feature.

Figure 13 The influence weight of different input features.

From the aforementioned analysis, it is evident that historical demand is the most crucial input feature. Therefore, in BDP model, the vector generated by inputting the historical demand graph feature into STGCN Block1 should be the vector with the highest association weight.

Model structure rationalization experiment

In order to verify the rationality of Self-Attention in ST-BDP model and analyze the influence of different feature inputs on the prediction performance, BDP model, TCN model, and two variants of BDP model (BDP-1 and BDP-2) are selected for performance comparison experiments. The four models are described as follows: BDP model: This model shares a structural similarity with ST-BDP model, being mainly bifurcated into the Encoder and Decoder segments. Distinctively, the BDP model does not incorporate demand graph features during its feature construction. Instead, it relies solely on the analysis of historical demand data.

TCN model: This model is mainly used for time series prediction, including time series prediction, causal convolution, inflated convolution, TCN residual block and other parts. Notably, it does not employ the Self-Attention mechanism, marking a difference from the BDP model.

BDP-1 model (without temporal feature): This model variation excludes the characteristic information of the time signal. Specifically, it omits the TCN Block2, which processes the input temporal feature in the original BDP model.

BDP-2 model (without weather feature): BDP-2 abstains from incorporating weather feature information. Within this model, the TCN Block 3, responsible for processing the weather factor, is removed from the BDP structure.

Self-attention performance impact analysis

When comparing TCN model to BDP model, a key distinction is the absence of the Self-Attention mechanism in TCN model. In Table 7, a detailed comparison between the prediction performance of the TCN and BDP models on the bike-sharing demand dataset is presented.

Table 7 Prediction performance of the TCN and BDP models.

Model	MAE	MAPE	SMAPE	RMSE	
TCN	2.11	19.93%	20.42%	3.21	
BDP	1.80	18.61%	18.82%	2.67	

By analyzing the experimental results in Table 7, it is found that the prediction performance of BDP model is better than that of TCN model, with lower MAE, MAPE, SMAPE, RMSE and higher accuracy. As the models’ prediction results shown in Fig. 14, BDP model’s predictions align more closely with the actual data compared to those from the TCN model. This superior performance of the BDP model demonstrates the efficacy of the Self-Attention mechanism in effectively capturing the long-term dependencies inherent in bike-sharing demand. By emphasizing relevant features and diminishing the impact of less pertinent ones, the Self-Attention mechanism enhances the model’s predictive accuracy.

Figure 14 An example demand prediction diagram for TCN model and BDP model.

Multi-spatio-temporal feature performance impact analysis

The performance comparison between the BDP model, BDP-1 model, and BDP-2 model, each having varied feature inputs, on the curated bike-sharing demand dataset, can be observed in Table 8.

Table 8 Prediction performance of BDP-1, BDP-2 and BDP models.

Model	MAE	MAPE	SMAPE	RMSE	
BDP-1	2.36	22.38%	22.92%	3.44	
BDP-2	3.07	22.88%	23.62%	5.49	
BDP	1.80	18.61%	18.82%	2.67	

Figure 15 illustrates the predictive performance of the BDP model and its two variants, bike-sharing demand prediction for the subsequent 72 h. BDP-1 model showcases a higher prediction accuracy compared to BDP-2 model. This suggests that weather features play a more substantial role in influencing prediction accuracy than the time signals. This is likely because the input historical demand data inherently functions as a time series, inherently capturing some of the temporal feature. However, the accuracy of BDP-1 model falls short when compared to the primary BDP model. This is attributed to the fact that the temporal features embedded within the historical demand series do not adequately capture the cyclic nature of bike-sharing demand. In contrast, explicit temporal signals provide a more comprehensive representation of this cyclic behavior.

Figure 15 An example diagram of BDP variant model and BDP model demand prediction.

Baseline comparison experiments

ST-BDP model’s predictive performance is benchmarked against an array of baseline models to understand its robustness and accuracy in bike-sharing demand prediction. These baseline models encompass traditional statistical methods, machine learning algorithms, and deep learning architectures: Historical average (HA): This model capitalizes on historical data, simply utilizing the average of previous periods as its predicted value.

Autoregressive Integrated Moving Average model (ARIMA): A stalwart in the time-series forecasting realm, ARIMA models data in a time series based on historical patterns. Its limitation lies in its inability to aptly fit irregular historical data. Furthermore, it primarily looks at time dependence, sidelining spatial network attributes, which could influence demand prediction.

Multilayer perceptron (MLP): A feedforward neural network that can solve nonlinear problems by stacking multiple layers of neurons and introducing nonlinear activation functions. However, for high-dimensional data, this network has high requirements for feature engineering and data preprocessing.

Recurrent neural network (RNN): This network model have internal states that allows them to remember contextual information from sequences, capturing dependencies effectively. This makes them well-suited for processing and predicting sequence data. However, RNNs struggle with capturing long-term dependencies in long sequences, often encountering issues such as vanishing or exploding gradients.

Long short-term memory (LSTM): This network model can effectively capture and retain long-term dependencies by introducing memory units and gating mechanisms. However, due to its time step dependency, this model is difficult to train in parallel.

Bidirectional LSTM (BiLSTM) (Schuster & Paliwal, 1997): Bidirectional LSTM can more accurately capture the bidirectional dependencies in the time series by processing the forward and backward information of the sequence simultaneously, thereby improving the prediction performance.

Gated recurrent unit (GRU): The GRU controls the writing of new information by merging the input gate and the forget gate into an update gate. However, it is difficult to perform well in complex prediction tasks because it is difficult to control the flow of information in a fine-grained manner.

Convolutional neural network (CNN): A deep learning model that can efficiently process grid-structured data through local connections, weight sharing, and hierarchical feature extraction. However, the network model is not robust enough to translation, rotation, and scale changes of the input data, and its effectiveness depends on the quality of data annotation.

Linear support vector regression (LSVR): A machine learning algorithm that is commonly used in binary classification problems, but can also be used to solve time-series prediction problems. It is mainly used to fit numerical values and is generally applied to scenarios with sparse features and a small number of features.

Feed-forward neural network (FNN) (Zhang et al., 2022): A basic neural network model with high applicability. The model can be considered as a function that implements a complex mapping from the input space to the output space through multiple compositions of simple nonlinear functions.

Graph convolutional GRU (GCGRU): A kind of graph convolutional neural network designed for temporal prediction with good prediction results, but the computation process is sequential computation of GRU.

Spatio-Temporal Graph Convolutional Networks (STGCN): A graph neural network improved variant model with spatio-temporal convolutional (TCN) blocks that can capture temporal and spatial features of the demand data, and the computation process is parallel. Compared with ST-BDP, the input features of STGCN do not introduce weather and periodic time features, and STGCN is not designed as an encoder-decoder model structure.

Table 9 shows the prediction performance of the ST-BDP model and the baseline model on the experimental dataset, where the time window of the LSTM model is 24 time steps, each time step is 1, and the grid search method is used to optimize the hyperparameter settings:

lr∈[0.001,0.0005,0.0001]

batch_size∈[16,32,64]

unit_number∈[64,128,256].

Table 9 Predictive performance of the baseline and ST-BDP models.

Model	MAE	MAPE	SMAPE	RMSE	
HA	5.89	40.45%	42.33%	10.97	
LSTM	5.35	32.94%	39.48%	10.82	
ARIMA	4.07	35.76%	38.21%	7.49	
LSVR	3.92	32.11%	35.73%	6.57	
MLP	3.82	31.14%	35.03%	6.50	
RNN	3.57	27.66%	32.05%	6.56	
GRU	3.47	26.02%	28.26%	6.62	
FNN	3.41	25.12%	27.46%	5.96	
CNN	3.34	25.78%	26.37%	6.13	
BiLSTM	2.92	23.91%	25.15%	5.21	
GCGRU	2.84	22.34%	24.81%	4.86	
STGCN	1.79	18.01%	18.24%	2.53	
ST-BDP	1.62	15.82%	16.14%	2.36	

The bidirectional LSTM model selects the same parameter settings as the LSTM model.

Through comparison of experimental results, it is found that the demand forecasting model based on deep learning (LSTM, RNN, GRU, GRU, CNN, GCGRU, STGCN, ST-BDP) has better prediction accuracy and lower prediction error rate than the demand forecasting model based on statistical learning methods (HA, ARIMA) and machine learning methods (MLP, LSVR, FNN).

When benchmarked against other models, graph neural network-based models (GCGRU, STGCN, ST-BDP) have excellent performance in demand forecasting. While traditional statistical learning methods and machine learning-based demand forecasting models can effectively address short-term forecasting challenges, their over-reliance on historical patterns, error accumulation, and insufficient spatial data integration affect the accuracy and effectiveness of their long-term forecasts. Although models such as LSTM, RNN, BiLSTM, and BDP can capture dependencies in time series data, they only rely on temporal features. Therefore, they cannot explain the complex spatial relationships and grid structures inherent in certain forecasting tasks, resulting in low forecasting accuracy. In contrast, the ST-BDP model built on the basic STGCN model adopts STGCN and TCN blocks to encapsulate additional features (demand graph, temporal signals, and weather information). This infusion of diverse feature information, especially the demand graph feature, renders the ST-BDP model exceptionally proficient, evidenced by its commendable performance across all four evaluation criteria: MAE, MAPE, SMAPE, and RMSE.

The prediction results of the ST-BDP model and other baseline models for the shared bicycle demand in the next 72 h are shown in Figs. 16 and 17. The baseline models shown in Fig. 17 are prediction models that only consider time characteristics. By observing the experimental indicators in the experimental results, STGCN model and ST-BDP model capture the trend of bike-sharing demand during the peak time period more accurately than the other models. This shows that the models based on efficient graph convolution and stacked temporal convolution structures, can quickly respond to dynamic demand changes in the bike-sharing network without overrelying on historical averages as most recurrent neural networks do.

Figure 16 Example demand prediction diagram for the baseline and ST-BDP models(a).

Figure 17 Example demand prediction diagram for the baseline and ST-BDP models(b).

In terms of model prediction efficiency, Fig. 18 shows the prediction time of the GCGRU model, STGCN model, and ST-BDP model on some data in the test set. In the experiment, the GCGRU model contains three layers, each with 64, 64, and 128 units. The setting of the STGCN model is basically the same as that of the ST-BDP model. The average prediction time of the three models on the test set obtained in the experiment is 0.571, 0.077, and 0.102s, respectively. Experimental results show that the prediction efficiency of the ST-BDP model is 5.6 times that of the GCGRU model. The ST-BDP model uses a graph convolutional network (GCN) to process spatial dependencies and a temporal convolutional network to process temporal dependencies. This model architecture allows the model to parallelize convolution operations in spatial and temporal dimensions, improving model efficiency. Compared with the STGCN model, the ST-BDP model adds components specifically for processing weather information and time signals, which makes the ST-BDP model structure more complex and slightly increases the prediction time. Thanks to the special design of ST-Conv Blocks and TCN Block, the ST-BDP model excels in capturing spatiotemporal features and processing multi-source data, while effectively balancing time consumption and parameter settings.

Figure 18 Comparison of single prediction time of GCGRU, STGCN and ST-BDP.

In short, the ST-BDP model exhibits superior precision in pinpointing the demand periods. This is because ST-BDP adds two TCN blocks to learn weather features and time periodic features in different dimensions. Furthermore, ST-BDP model is architectured with an encoder-decoder design that incorporates Self-Attention mechanism, enhancing the model’s efficacy even more.

Discussions

Experimental results demonstrate that our proposed ST-BDP model exhibits superior accuracy in predicting shared bikes demand while maintaining computational efficiency. Compared to the STGCN model, the ST-BDP model not only employs graph convolution and stacked temporal convolution structures but also integrates additional features such as demand graphs, time signals, and weather information. The inclusion of weather information allows the ST-BDP model to adapt to demand fluctuations under various weather conditions. The processing of time signals enhances the model’s ability to adapt to periodic fluctuations and special time period demand changes in shared bikes. Furthermore, the fusion of demand network characteristics enables the model to perceive spatial heterogeneity, thereby more accurately predicting demand interactions and flows between different regions and improving the prediction accuracy for different time periods. In addition, compared to traditional models such as RNN, the ST-BDP model utilizes ST-Conv Blocks and TCN Block structures, efficiently capturing spatiotemporal features through parallel processing, which provides significant advantages in computational efficiency. The accurate demand predictions from the ST-BDP model aid in optimizing resource allocation, reducing idle bicycles, and enhancing bicycle utilization. Moreover, the model not only ensures high prediction accuracy but also optimizes computational efficiency, improving its practicality for real-time scheduling and dynamic optimization scenarios. This capability supports shared bikes operators in quickly responding during peak periods and emergencies.

Conclusions

This article proposes a bike-sharing demand prediction model based on spatio-temporal graph convolutional networks, called ST-BDP model. Within ST-BDP, a STGCN and two TCN blocks play a pivotal role in extracting spatio-temporal feature form demand graph, weather and temporal characteristics. The encoder-decoder structure based on Self-Attention mechanism is used to facilitate hyperparameter learning of ST-BDP and output of prediction results.

The efficacy of ST-BDP model was rigorously assessed using a series of experiments, encompassing feature construction, model structural analysis, and comparisons with baseline models. These evaluations were conducted on a real-world dataset from New York, USA. The empirical results highlight that ST-BDP outperforms the baseline models in terms of predictive accuracy while ensuring reduced training overheads. Specifically, the model demonstrates a robust capability to accurately predict the daily trends in bike-sharing demand, all the while maintaining computational efficiency. These findings substantiate the judicious choice of incorporating temporal, weather, and demand characteristics as feature inputs, and further underscore the structure superiority of ST-BDP model. Although the ST-BDP model performs well in bike-sharing demand forecasting, it still has some limitations. For example, when forecasting demand for shared bikes, the model failed to fully consider special circumstances, which may result in the forecast model being less accurate in some specific situations. In addition, the model’s predictive capabilities rely on patterns in historical data, and when demand patterns change significantly, the model may have difficulty adapting quickly.

In future endeavors, we plan to use online learning methods to enable the model to update and adapt to new data patterns in real time. At the same time, we aim to devise a delivery strategy program rooted in the predictive outcomes of our bike-sharing demand model (ST-BDP). Our aspirations extend to a comprehensive exploration of dynamic transportation resource allocation, specifically for the route planning associated with bike-sharing logistics. Our overarching objective is to introduce technological solutions that not only curtail operational costs of transportation resources but also enhance the commuting experience for urban dwellers, all while aligning with urban traffic optimization goals.

Additional Information and Declarations

Competing Interests

Author Contributions

Data Availability

The authors declare that they have no competing interests.

Chaoran Zhou conceived and designed the experiments, performed the experiments, analyzed the data, performed the computation work, prepared figures and/or tables, authored or reviewed drafts of the article, and approved the final draft.

Jiahao Hu conceived and designed the experiments, performed the experiments, analyzed the data, performed the computation work, prepared figures and/or tables, and approved the final draft.

Xin Zhang conceived and designed the experiments, prepared figures and/or tables, authored or reviewed drafts of the article, and approved the final draft.

Zerui Li performed the experiments, prepared figures and/or tables, and approved the final draft.

Kaicheng Yang analyzed the data, prepared figures and/or tables, and approved the final draft.

The following information was supplied regarding data availability:

The Citi Bike System Data is available at Citibike and Gitee:

- https://citibikenyc.com/system-data

- https://gitee.com/jiahaohuhu/sharebike/blob/main/Experiment_01/Dataset/bike_sequences_one_hot.npz

The code is available at Zenodo and Gitee:

- Hu, J. (2024). The code of Bike-sharing Demand Prediction model based on Spatio-Temporal Graph Convolutional Networks. Zenodo. https://doi.org/10.5281/zenodo.13160657

- https://gitee.com/jiahaohuhu/sharebike/tree/main

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
