# Peer review of "A bike-sharing demand prediction model based on Spatio-Temporal Graph Convolutional Networks"

_PeerJ Computer Science, doi:10.7717/peerj-cs.2391_

## Round 0.1 · original submission · Major Revisions

Dear Authors,

Please, carefully follow the suggestion provided by reviewer 1 and reviewer 3.

Reviewer 1 ·

Basic reporting

The authors propose a novel AI model architecture for the problem of Bike-sharing Demand Prediction.
The introduction effectively describes the context of data-driven solutions regarding Bike-sharing systems and highlights the challenges to be addressed. The related works section describes the literature on Bike-sharing data-driven solutions contextualizing these works in the broader field of traffic flow. This could be appropriate in the broader field of time-series analysis and predictions, however, I suggest the authors to put in a table the reported works on related Bike-sharing metrics with details on the results in terms of metrics achieved and the details of the designed models.
The structure of the article needs to be improved. At the current state, the authors mix the ST-BDP model presentation followed by some data feature details, for then detailing the model components and then providing details of the dataset construction. I suggest the authors to present the dataset details first followed by a complete section on the structure of the model proposed to improve clarity. I suggest also to inlcude a paper structure section at the end of the introduction helping to have an overview of the different parts.

Experimental design

The experimental section of the paper is well described and the findings are relevant. The proposed solution fills the challenges stated in the introduction exploiting not only multivariate temporal features but also the spatial information formulating the problem with also a graph representation of the bike-sharing system. The proposed ST-BDP model outperforms baseline models in terms of results on the test set on adequate evaluation error metrics. However, based on the related works this comparison should include also deep learning architectures that exploit only temporal features such as a deep bidirectional Long Short-Term Memory (LSTM) network and compare the results.

Validity of the findings

The authors proposed a novel AI model architecture for the Bike-sharing demand prediction that exploits not only temporal features but also spatial information (ST-BDP). This model has been benchmarked with other baseline models and demonstrated superior performance in demand prediction. The authors stated in the conclusion that this model execution is also computationally efficient due to its architecture. I suggest the authors in the comparison table to add also the average execution time on the test set per single prediction to demonstrate the computational efficiency

Additional comments

The authors should check the text for some missings such as:
- line 210 probably missing σ
- line 273 Block1 and Block2 should be Block2 and Block3

Reviewer 2 ·

Basic reporting

Very well written but there are typos and needs to be reviewed.
Literature references are up to date but need to be reviewed in the text as there are double references.

Experimental design

No comment

Validity of the findings

No comment

·

Basic reporting

Review the manuscript for minor grammatical errors and awkward phrasings to improve readability. Consider using a professional editing service for language polishing.

Include a more detailed comparison of the proposed model with existing models in the field, highlighting the improvements and novel aspects.

Expand the Discussion section to include more detailed analysis and interpretation of the results, discussing the implications and potential applications of the findings.

Provide more detailed explanations of the mathematical formulations and theorems used in the model.

Experimental design

Provide more detailed descriptions of the model's architecture and training process to enhance clarity and replicability.

Validity of the findings

The impact and novelty of the findings are implied but not explicitly assessed. A more explicit discussion of the potential impact and novelty of the model would strengthen the article.Include a discussion on the impact and novelty of the findings to highlight the significance of the research. The underlying data provided are robust, statistically sound, and well-controlled. The experimental design and data analysis are appropriate for the research question.The conclusions are well stated and directly linked to the original research question. They are appropriately limited to the supporting results. Expand the conclusions to include a discussion on potential limitations of the study and suggestions for future research.

Additional comments

The article presents a significant contribution to the field of bike-sharing demand prediction. The use of spatio-temporal graph convolutional networks is innovative and well-executed.
The Discussion section should be expanded to include more interpretation of the results, potential limitations of the study, and suggestions for future research.
Minor grammatical and typographical errors should be corrected to improve readability.

Compare the study with the results in the following papers

Sathishkumar, V. E., Jangwoo Park, and Yongyun Cho. "Using data mining techniques for bike sharing demand prediction in metropolitan city." Computer Communications 153 (2020): 353-366.

VE, S., & Cho, Y. (2020). A rule-based model for Seoul Bike sharing demand prediction using weather data. European Journal of Remote Sensing, 53(sup1), 166-183.

VE, S., & Cho, Y. (2024). Season wise bike sharing demand analysis using random forest algorithm. Computational Intelligence, 40(1), e12287.

Cite this review as

---

## Round 0.2 · Major Revisions

Dear Authors,

Please, go through all the points raised by reviewer #1 and carefully answer.

Reviewer 1 ·

Basic reporting

The authors assessed the requirements suggested regarding paper structure. Although the authors included a table of related works, it did not fully meet the expectations for reporting only bike-sharing related metrics, as it included works such as the one by Zhao et al. (2022), which is related to traffic. Furthermore, I suggest clearly reporting the performance results obtained through metrics, rather than relying on qualitative impressions such as "the three models all work well" and "the most suitable."

Experimental design

The authors did not fully address the requirement to compare the solution with a deep bidirectional Long Short-Term Memory network.
However, the comparison reports an LSTM network that utilizes only temporal features, which is what I was asking to be explored.
I suggest including details about the temporal window used for the LSTM architecture to better understand the experiments performed. Additionally, please report if any hyperparametrization procedure was performed on this architecture. These suggestions arise from the results reported in Table 9, which need clarification
Model MAE MAPE SMAPE RMSE
LSTM 5.35 27.66% 32.05% 6.56
ARIMA 4.07 35.76% 38.21% 7.49
RNN 3.57 27.66% 32.05% 6.56

It is unusual that the MAE for the LSTM is higher than that for ARIMA, while other metrics show a lower error for the LSTM. This discrepancy may be due to a reporting error, particularly since the RNN results appear to match those of the LSTM.

Validity of the findings

The authors reported the time required to train the network, but I requested information on the time required for network execution. Specifically, reporting the average execution time per single prediction on the test set would provide a clearer demonstration of the model's computational efficiency.

·

Basic reporting

Good

Experimental design

Well done

Validity of the findings

Good

Additional comments

NA

Cite this review as

---

## Round 0.3 · accepted · Accept

Dear Authors,

According to the reviewer(s) you answered all their questions improving the quality of the manuscript.

M.P.

Reviewer 1 ·

Basic reporting

The authors improved the related works accordingly to the provided suggestions.

Experimental design

The authors, based on the suggestions, reported the details requested and improved the overall model comparison section.

Validity of the findings

The authors improved the model's computational execution time as requested validating the efficiency of the proposed solution (<1s).

Additional comments

The authors have improved the manuscript, based on the suggestions proposed.